# Molecular Threat of Splicing Factor Mutations to Myeloid Malignancies and Potential Therapeutic Modulations

**DOI:** 10.3390/biomedicines10081972

**Published:** 2022-08-15

**Authors:** Fangliang Zhang, Liang Chen

**Affiliations:** Hubei Key Laboratory of Cell Homeostasis, RNA Institute, College of Life Sciences, Wuhan University, Wuhan 430072, China

**Keywords:** myelodysplastic syndromes, acute myeloid leukemia, spliceosome, DNA damage response, immune response pathway

## Abstract

Splicing factors are frequently mutated in myelodysplastic syndromes (MDS) and acute myeloid leukemia (AML). These mutations are presumed to contribute to oncogenic transformation, but the underlying mechanisms remain incompletely understood. While no specific treatment option is available for MDS/AML patients with spliceosome mutations, novel targeting strategies are actively explored, leading to clinical trials of small molecule inhibitors that target the spliceosome, DNA damage response pathway, and immune response pathway. Here, we review recent progress in mechanistic understanding of splicing factor mutations promoting disease progression and summarize potential therapeutic strategies, which, if successful, would provide clinical benefit to patients carrying splicing factor mutations.

## 1. Introduction

Splicing factors are frequently mutated in a variety of hematologic malignancies [1,2,3], including myelodysplastic syndromes (MDS) that may progress to acute myeloid leukemia (AML) [4,5,6]. Among all splicing factor mutations, major heterozygous point mutations in Splicing factor 3B subunit 1 (*SF3B1*), U2 small nuclear RNA auxiliary factor 1 (*U2AF1*), Serine and arginine rich splicing factor 2 (*SRSF2*), and RNA binding motif and serine/arginine rich 2 (*ZRSR2*) have been demonstrated as driver mutations to promote disease progression [2,7]. RNA sequencing studies further revealed that all of these mutations altered spliceosome functions, resulting in numerous sequence-specific mis-splicing events [8,9,10,11]. Interestingly, mutations in different splicing factors are mutually exclusive and only heterozygous mutations were discovered in patients [12]. These observations suggest that mutations in different splicing factors may have convergent adverse effects beyond cell tolerance and mutant cells rely on the canonical functions of the wild-type allele for survival [2]. Indeed, recent studies using CRISPR/Cas9 to selectively knock out the wild-type allele of the splicing factor genes confirmed the dependency of mutation-carrying cells on the wild-type allele [12,13].

Mutations in *SF3B1*, *U2AF1*, and *SRSF2* have also been shown to promote aberrant R-loop formation, which triggers replication stress, followed by DNA-damage responses, specifically the activation of Ataxia telangectasia and Rad3 related (ATR)-CHK1 [14,15,16]. These mutations may elicit innate immune pathways as well, leading to chronic activation of the inflammasome and inflammatory cytokine production [17,18]. All these findings of diverse disease mechanisms have raised a broad interest in developing new therapeutical strategies to modulate splicing regulatory activity, block specific DNA damage repair pathways and target immune response and inflammatory pathways. Here, we summarize recent progress toward an understanding of the pathogenic roles of splicing factor mutations and the development of new targeted therapies with small molecules.

## 2. The Discovery and Pathogenic Roles of Splicing Factor Mutations during the Development of MDS/AML

Efforts in the last decade have revealed that mutations in genes encoding RNA splicing factors are commonly present in patients with myeloid neoplasms (MN) [2], chronic lymphocytic leukemia (CLL) [19], clonal hematopoiesis [20], and mantle cell lymphoma [21]. MDS are clonal stem cell malignancies characterized by ineffective hematopoiesis, leading to peripheral cytopenia and a variable risk of progression to AML, one of the most deadly blood cancers. The ineffective hematopoiesis is largely attributed to the dysregulated hematopoietic stem cells (HSCs), which are critical for sustained production of all types of blood cells throughout the lifetime [22]. Through a delicate balance between self-renewal, quiescence, and differentiation, HSCs are maintained under homeostatic conditions, yet would dynamically respond to all kinds of intracellular/extracellular stimuli [23,24].

The spliceosome complex is the principal machinery to catalyze RNA splicing reactions. Within the spliceosome, there are five small nuclear ribonucleoproteins (snRNPs) that contain specific snRNAs, including U1, U2, U4, U5, and U6, and more than 300 related proteins [17,25,26]. The major function of the spliceosome is to remove introns and ligate exons of a pre-mRNA in different ways by favoring or repressing the use of various splice sites, resulting in the production of different mRNA isoforms [27]. Early studies by next-generation sequencing revealed a large number of aberrant alternative splicing events in MDS [28], and over 50% of MDS patients have mutations in splicing factors [29], such as serine and arginine domain containing factors, including *SRSF2*, *U2AF1*, and *ZRSR2*, and a U2 snRNP component *SF3B1* (in Table 1). These defects are thought to occur early in the disease process [5,30], causing changes in cellular pathways involved in hematopoiesis, DNA damage response, epigenetic modifications, immune responses, and iron metabolism through mis-splicing or nonsense-mediated decay [9,17,18,31,32]. Combined with other defects in the regulation of transcription, replication, and genome instability, these hotspot mutations interfere with normal hematopoiesis and induce malignancies.

Mutations in *SF3B1* are detected across various types of blood and solid tumors, such as CLL [3,19,43] and uveal melanoma (UM) [44]. The mutation frequency is highest in MDS and particularly in MDS with ring sideroblasts (MDS-RS) [1]; therefore, *SF3B1* mutations are now used to define a distinct clinical entity of MDS-RS [2,45]. Moreover, *SF3B1* mutations often occur as heterozygous point mutations at restricted residues within its HEAT domain, including E622, R625, H662, K666, and K700 [2,3], resulting in the use of cryptic 3′ splice site (ss) located at −15 to −24 nt of the canonical 3′ ss with short polypyrimidine tracts. Mechanistically, this is due to the preference of mutant *SF3B1* to use an alternative branch-point Adenosine motif [8,46]. Splicing changes associated with *SF3B1* mutations have been found in many tumor-related genes, including Neurofibromin 1 (*NF1*), Dicer 1-ribonuclease III (*DICER1*), Promyelocytic leukemia protein (*PML*), Cohesin associated factor A (*PDS5A*), Mitogen-activated protein kinase kinase kinase 7 (*MAP3K7*), Protein phosphatase 2 regulatory subunit B’alpha (*PPP2R5A*), and Bromodomain containing 9 (*BRD9*) [37,47]. Many aberrant transcripts are expected to generate premature stop codons subject to nonsense-mediated decay.

Regarding the clinical significance of *SF3B1* mutations, early studies suggested that MDS with *SF3B1* mutations were associated with a better prognosis and a lower likelihood to transform into AML compared with MDS free of *SF3B1* mutations [48,49,50,51,52]. MDS and UM patients carrying *SF3B1* mutations also appeared to have better survival [53,54]. In contrast, these conclusions were not reproduced in some other studies [55,56], especially when patients harbor other gene mutations [48]. *SF3B1* mutations in UM were later linked to a higher risk of metastasis as well [57]. Interestingly, the *SF3B1 K700E* mutation promotes alternative splicing of Interleukin-1 receptor-associated kinase 4 (*IRAK4*) to generate a long isoform (*IRAK4-L*) in MDS and AML, which interacts with MyD88 to elicit hyperactivation of NF-κB and leukemogenesis [10,38]. Clinically, the expression of *IRAK-L* is associated with a poor prognosis of MDS and AML [17]. Other tumorigenic mechanisms, including MYC stabilization, were also reported [8,58]. Thus, the prognostic relevance of *SF3B1* mutations in MDS and other diseases may depend on the cellular context and await further confirmation.

U2AF1 and U2AF2 constitute the U2AF heterodimeric complex critical for delineating the 3′ ss. Mutations in *U2AF1* are restricted to myeloid neoplasms, especially associated with high-risk MDS (HR-MDS) and AML [4,7]. There are two mutation hotspots in *U2AF1*, the S34 and Q157 residues located within the first and second zinc-finger domains [2,7]. U2AF1 recognizes the consensus AG dinucleotide of the 3′ss, yet the mutant protein changes recognition preference for bases flanking the AG nucleotide, causing altered 3′ss selection, and aberrant exon skipping and inclusion [59]. Intriguingly, S34 and Q157 mutations were shown to alter the splicing of distinct groups of genes, suggesting a site-specific effect on protein function [11,31].

Genes with altered splicing patterns in *U2AF1* mutant cells are involved in several major biological pathways, including (1) epigenetic regulation: H2A histone family member Y (*H2AFY*), ASXL transcriptional regulator 1 (*ASXL1*), BCL6 corepressor (*BCOR*), and DNA methyltransferase 3 beta (*DNMT3B*); (2) apoptosis: Caspase 8 (*CASP8*); (3) DNA damage response: *ATR* and FA complementation group A (*FANCA*) and (4) innate immune signaling: *IRAK4* [11,17,31]. A few other genes, the altered splicing of which possibly contributes to transformation into malignant cells include Serine/threonine kinase receptor associated protein (*STRAP*), Centrosomal protein 164 (*CEP164*), Euchromatic histone lysine methyltransferase 1 (*EHMT1*), WW domain containing adaptor with coiled-coil (*WAC*), Poly (A) binding protein cytoplasmic 4 (*PABPC4*), Peptidylprolyl isomerase domain and WD repeat containing 1 (*PPWD1*), Polypyrimidine tract binding protein 1 (*PTBP1*), and UPF3B regulator of nonsense mediated mRNA decay (*UPF3B*) [40]. Of note, *U2AF1* mutations also elicit the expression of *IRAK4-L* in the majority of MDS patients.

Mutations in *SRSF2* are related to the HR-MDS and AML as well [4,7]. About 10% of AML, 20% of MDS, and 50% of patients with chronic myelomonocytic leukemia (CMML) harbor *SRSF2* mutations [2,4,60]. The most frequently occurred mutation is P95H, which may markedly change the RNA binding specificity of SRSF2, thereby altering the recognition of exonic splicing enhancer motifs from GGNG to CCNG [9,61]. Many genes differentially spliced by mutant *SRSF2* are involved in myeloid malignancies. For example, mutant *SRSF2* promotes inclusion of a highly conserved poison exon within the Enhancers of zeste homologue 2 (*EZH2*) transcript and triggers its nonsense-mediated decay, which might lead to impaired differentiation of mutant hematopoietic cells [9]. It is worth noting that *EZH2* loss-of-function mutations are common in MDS and mutually exclusive with *SRSF2* mutations [4]. Moreover, P95H mutation represses the inclusion of a cassette exon in *CASP8*, resulting in a truncated protein that lacks the C-terminal catalytic domains and induced NF-κB signaling [42].

ZRSR2 is primarily involved in the function of the minor spliceosome [35], and has been proved as a regulator of hematopoiesis [33]. The gene encoding ZRSR2 is located on the X chromosome and frequently harbors nonsense or frameshift mutations in MDS patients, resulting in the inactivation of gene function in male patients [2]. Loss of *ZRSR2* impairs minor (U12) intron excision and enhances HSC self-renewal by increasing the expression of the Leucine zipper like transcription regulator 1 (*LZTR1*) and a regulator of Ras-related guanosine triphosphate hydrolyzing enzymes (GTPases) [33]. Mutations in *ZRSR2* and the U2AF heterodimer have been found in approximately 5% of MDS patients [35].

GO analyses of mis-spliced genes in MDS patients with *ZRSR2* mutations found that the expression of MAPK pathway members (*MAPK1, MAPK3*), E2F transcription factors (*E2F1, E2F2, E2F3, E2F4, E2F6*), RAS guanyl releasing protein (*RASGRP1, RASGRP2, RASGRP4*), RAF serine/threonine protein kinases (*ARAF, BRAF, RAF1*), and the tumor-suppressor gene *PTEN* were mainly affected [35]. *Zrsr1* is a closely related homolog of *Zrsr2* expressed in murine hematopoietic cells, but not in human cells. Depletion of *Zrsr1* in *Zrsr2* KO myeloid cells exacerbated retention of the U12-type introns of *MAPK9* and *MAPK14* and further reduced protein expression [36]. Blastic plasmacytoid dendritic cell neoplasm is an aggressive leukemia of plasmacytoid dendritic cells known with male predominance. It was recently shown that *ZRSR2* mutations impaired plasmacytoid dendritic cell activation and enhanced apoptosis after inflammatory stimuli, which may be attributed to intron retention and therefore failed upregulation of the transcription factor *IRF7*, a downstream Toll-like receptor, in response to inflammatory signals [34]. Altogether, ZRSR2 may play an integral role in hematopoietic development and mutations may thus be linked to leukemogenesis.

Moreover, mutations in other proteins and non-coding RNAs in splicing regulation with lower frequency have been shown to play important roles in myeloid neoplasms, too. Mutations in genes encoding the U1 and U11 snRNAs responsible for recognition of the 5′ss in the major and minor spliceosomes were found in CLL and other cancer types [62,63]. Factors in the hnRNP family function to repress splicing as opposed to SR proteins [21], and mutations in *HNRNPH1* are found to be associated with adverse outcomes in mantle cell lymphoma [64]. Mutations in other splicing factors were discovered in MN with lower frequencies as well, including *PRPF8* mutations in 3% of MN patients [65] and *DDX41* mutations in 0.5% to 5% of MDS/ AML cases [66,67]. The RNA helicase *DDX41* is involved in the regulation of RNA splicing, snoRNA processing, and ribosomal RNA biogenesis [68,69]. Recent studies have reported that *DDX41* mutations in MN are associated with longer overall survival and response to lenalidomide [66,69,70,71].

## 3. Mechanisms That Contribute to Leukemogenesis beyond Splicing Defects

HSCs may suffer from environmental or therapy-related challenges, and endogenous transcriptional, replicative, and oxidative stress, all of which could threaten genome integrity [72], exerting selective pressure on preleukemic/clonal hematopoiesis of indeterminate potential (CHIP) clones and driving cancer progression [15,73,74]. Strikingly, many somatic mutations seen in MDS/AML can be detected in untransformed hematopoietic stem-progenitor cells (HSPCs) of healthy persons [75]. Along with HSPCs acquiring initiating mutations, non-pathogenic mutations are carried during clonal expansion. These preleukemic cells may survive treatment, accumulate additional mutations, become the origin of malignancies [75,76,77], and eventually contribute to relapse [76]. Therefore, it is important to understand the regulation and roles of various DNA damage responses (DDR) in HSCs/leukemic stem cells (LSCs), which constitute an intricate system for genomic integrity maintenance and prevention of mutated sites accumulation [78].

Interestingly, MDS-associated *SRSF2* and *U2AF1* mutations have been shown to enhance R-loop formation as a consequence of impaired transcription pause release, independent of splicing defects [15]. R-loop is a three-stranded nucleic acid structure composed of an RNA strand hybridized with one strand of DNA through sequence complementarity, leaving the other strand displaced [16]. Such a unique structure has been shown to regulate transcription and epigenetic functions, yet if unresolved properly, may cause conflict between transcription and DNA replication. Consistently, aberrant R-loop regulation in mutant cells subsequently triggers replication stress and ATR-CHK1 activation when cells enter the S phase. Suppression of R-loops by overexpression of RNASEH1 partially rescues the compromised phenotype observed in *Srsf2* P95H-mutant hematopoietic progenitor cells [15]. Similarly, *SF3B1* mutations in K562 cells and MDS HSPCs also lead to DNA damage and activation of the ATR pathway, which can be partially rescued through resolving R-loops [14]. Importantly, such disease mechanism appears to be universal in certain non-splicing factor mutant cells as well, such as R-loop-driven MDS associated with BRCA1 haploinsufficiency [79]. On the other hand, as mentioned above, splicing of certain genes involved in DDR and cell cycle control are selectively vulnerable to splicing factor mutations in MDS and AML [80,81], thus suggesting multiple mechanisms may induce genome instability and drive blood disorders in response to splicing factor mutations.

MDS patients often have aberrant levels of chemokines and growth factors in the peripheral blood and bone marrow [82]. Specifically, the levels of TNF-α, IFN-γ, TGF-β, IL-6, and IL-8 are often higher in MDS patients relative to normal donors, and associated with both dysregulated inflammatory signaling and myeloid differentiation-block [82,83]. It has been suggested that the level of apoptosis in the bone marrow is inversely correlated with that of clonal expansion and MDS risk [84]. High levels of IFN-γ and IL-6 secretion in the bone marrow of MDS patients is generally related to induction of apoptosis and low-risk MDS (LR-MDS). In contrast, immunosuppressive cytokines, such as IL-10, are highly secreted in HR-MDS [82]. These correlations imply that immune abnormalities and stress may play important roles in disease progression from LR-MDS to HR-MDS or to AML [85].

Strikingly, mutations in *SF3B1*, *U2AF1*, and *SRSF2* seem to alter the activity of innate immune pathways and enhance chronic inflammation. These mutations were shown to enhance NF-κB activity and LPS-induced inflammatory cytokine production in macrophages, patient-derived cell lines, and mouse and human myeloid cells [17,18]. While *SF3B1* and *SRSF2* mutations elicit distinct effects on splicing, they are synthetically lethal, possibly due to the convergent impact on NF-κB activity, which subsequently affects the fitness and functions of hematopoietic stem cell [17,42,86].

## 4. Development of Drugs Targeting Cancer Cells with Splicing Factor Mutations

### 4.1. Direct Targeting of the Spliceosome with Splicing Factor Modulators

As the splicing defect is a direct consequence of mutations in splicing factors, predominant interests have been given to search for key downstream targets of mis-spliced genes in MDS/AML patients. Moreover, as MDS clones harboring splicing factor mutations may be preferentially sensitive to inhibition of the RNA splicing process compared to the unmutated stem/progenitor compartment [30], several drugs targeting the splicing machinery are currently evaluated in clinical trials. In particular, a variety of natural products and their derivatives that bind to the SF3B complex and inhibit pre-mRNA splicing at an early step of spliceosome assembly were discovered, including pladienolides, E7107, FR901464, spliceostatin A, herboxidiene, and sudemycin D6 (in Table 2) [87,88,89,90,91]. Extensive efforts have been made in the last few years to examine if these small chemicals have antitumor activity in MDS/AML patients [87,88].

Pladienolide is a natural macrolide [87], and E7107 is a semisynthetic derivative of pladienolide. Treatment of isogenic murine myeloid leukemias with E7107 revealed preferential death of leukemic cells bearing mutated *SRSF2* [12,96]. However, further clinical trials of E7107 were halted due to dose-limiting toxicity as participants developed vision loss without clinical benefit [109,110]. FR901464 displays strong antitumor activity and inhibits cell cycle progression at G1 and G2/M phases. However, it exhibits strong cytotoxicity at low nanomolar concentrations as well [92]. In a recent report, the synthetic derivative of pladienolide H3B-8800 appears to hold more promise for clinical application. H3B-8800 has exhibited good binding activity to spliceosomal components in a dose-dependent manner, and is safe even with prolonged dosing in myeloid neoplasms [94]. It is orally bioavailable and inhibits tumors with mutations not only in *SF3B1*, but also in *U2AF1* and *SRSF2* in xenograft leukemia models [3,93]. A related Phase I clinical trial (NCT02841540) is currently performed in patients with MDS, AML, or CMML [94,95], to determine its safety and potential therapeutic efficacy.

Spliceostatin A is a methyl ketal derivative of FR901464 and inhibits both splicing and nuclear retention of pre-mRNA [88]. It interferes with spliceosome assembly at the step following the binding of U2 snRNP with pre-mRNA [111]. Recently, spliceostatin A has been found to elicit apoptosis of chronic lymphocytic leukemia cells through downregulation of the pro-survival Bcl-2 family member Mcl-1, the high expression of which is correlated with disease progression [112]. Herboxidiene is a microbial product with antitumor activity, which serves as a novel splicing inhibitor that specifically impairs the function of SF3B by binding to SAP155 [90]. Sudemycin D6 is a small molecule that targets SF3B1 and inhibits spliceosome activity as well [91]. Hematopoietic cells expressing *U2AF1 (S34F)* are sensitive to sudemycin D6 [113], and treatment of *U2af1 (S34F)* transgenic mice with sudemycin D6 results in attenuated expansion and increased apoptosis of hematopoietic progenitor cells [114].

In addition, Rbm39 modulators bind to Rbm39 and recruit the E3 ubiquitin ligase CUL4-DCAF15 for Rbm39 degradation, leading to the disruption of the SF3B1 complex formation [97]. Protein arginine methyltransferase 5 (PRMT5) and Protein arginine methyltransferase 1 (PRMT1) methylate spliceosome components are required for spliceosome formation. Protein arginine methyltransferase 5/1 inhibitors reduce spliceosome methylation and inhibit splicing, and leukemias with splicing factor mutations are preferentially sensitive to PRMT5 inhibition [30,98,99]. A few PRMT5 small-molecule inhibitors, GSK3326595, JNJ-64619178, and PRT543, are currently in early-stage clinical trials to evaluate the specificity and efficacy in multiple tumors, including MDS and AML (NCT03614728, NCT03573310, NCT03886831), with the hope to benefit patients in the future.

In theory, splicing inhibitors are expected to possess limited selectivity for tumor versus normal cells. Therefore, the side effect is a potential issue for clinical use of spliceosome inhibitors. Toxicity might be alleviated by lowering splicing modulators’ doses at the expense of therapeutic efficacy. In any case, more work is needed to improve treatment expectations.

### 4.2. Targeting DDR with Small Molecule Inhibitors

On one hand, mis-repaired DNA damage promotes genomic instability and mutation accumulation, inducing cell transformation and cancer progression. On the other hand, DNA damage checkpoints can induce apoptosis, senescence, or differentiation, leading to depletion of stem cells and, ultimately, bone marrow failure [72,115]. Chronic DNA damage is accompanied by constitutive activation of checkpoint pathways in primary AML cells with complex karyotypes [116]. One interesting hypothesis is that hematopoietic cells with splicing factor mutations suffer cellular stress in both gene expression and replication, resulting in defective self-renewal and differentiation potential. Some mutant cells may undergo specific transformation and adaptation, and survive the stress, yet become selectively dependent on endogenous DNA damage repair activity at MDS/AML stages.

ATR is a serine/threonine-specific protein kinase that is activated in response to persistent single-stranded DNA, leading to activation of the DNA damage checkpoint and cell-cycle arrest. Previous studies show that malignant cells, including AML samples, are particularly sensitive to hypomorphic inhibition of ATR compared to normal tissues [117]. At the molecular level, ATR blockade results in G2/M checkpoint abrogation, DNA damage, and apoptosis in AML cell lines and primary tumor samples [118]. Interestingly, it has been shown that the transcript of ATR is alternatively spliced in *U2AF1*-mutant AML patient samples and the mutated *U2AF1* CD34 cells [31,119]. Consistently, cells with *U2AF1* or *SRSF2* mutation are more sensitive to ATR inhibition in vitro [15,74]. AZD6738 is an orally active ATR inhibitor and sensitizes p53- or ATM-defective primary CLL cells to chemotherapy and ibrutinib both in vitro and in vivo assays [120]. MDS CD34+ cells with splicing factor mutations (*SF3B1*, *SRSF2,* and *U2AF1*) are hypersensitive to ATR inhibitors (VE-821 and AZD6738) [73]. Encouraged by these pre-clinical results, AZD6738 is currently in the phase I trial in the treatment of MDS or CMML patients with at least one mutation in splicing factors (*SF3B1*, *U2AF1*, *SRSF2*, and *ZRSR2*), for safety and tolerability test (NCT03770429) [74,100].

While monotherapy studies aim to exploit specific molecular vulnerabilities of tumors, most clinical trials of ATR inhibitors (ATRi) are performed in combination with other drugs. The combination of ATRi and antimetabolite chemotherapy has been studied in the context of acute myeloid leukemia (AML); ATR inhibition by VE-822 treatment can enhance hydroxyurea- and gemcitabine-induced growth inhibition through S-phase cell cycle arrest and increased replication fork stalling in AML samples ex vivo, and VE-822 potentiates the cytotoxicity of gemcitabine in an orthotopic mouse model of AML [101]. Moreover, ATRi in combination with splicing modulation show augmented efficacy in CD34+ cells with splicing factor mutations [73].

As the downstream effector of ATRi, CHK1 is serine/threonine kinase that promotes cycle arrest in response to DNA damage or replication stalling/stress [121]. A CHK1 inhibitor (CHK1i) UCN-01 can selectively induce apoptosis of cells with s *SF3B1* mutation through cell cycle arrest, and increase the efficacy of the Sudemycin D6 when used together in the treatment of MDS/AML [14]. The nucleoside analog cytarabine is a potent and widely used drug in AML [122]. Etoposide inhibits topoisomerase II and induces cell cycle arrest, apoptosis, and autophagy [123]. Another small chemical SCH900776 was shown to disrupt cytarabine-induced CHK1 activation, leading to S phase arrest and significantly increased apoptosis in AML cell lines [104]. CHK1i also enhanced the effect of CPX-351 (liposomally encapsulated combination of cytarabine and daunorubicin) in AML cell lines and primary AML cells [124]. Moreover, the combination of ATRi and CHK1i shows synergistic suppression of ex vivo AML cell proliferation [103,125]. The elevated expression of cytosine deaminase APOBEC3A sensitizes AML cells to VE-822 and CHK1i of PF477736, thereby could be used as a biomarker for ATRi/CHK1i therapy of AML [102]. Additionally, combined inhibition of CHK1 and BCL-2 by LY2603618 and venetoclax synergistically induced apoptosis in AML cell lines and primary patient samples [105].

The tyrosine kinase Wee1 is a dual specificity kinase that phosphorylates CDK1 at Tyr15 to inhibit its kinase activity [126]. In response to DNA damage in the G2 phase, CHK1 phosphorylates the kinase WEE1 and the phosphatase CDC25C, resulting in CDK1 inhibition and cell cycle arrest at the G2/M checkpoint [121,126]. Combinatory inhibition of WEE1 and ATR/CHK1 showed a synergistic inhibitory effect in AML cells ex vivo [103,125]. In AML patient samples with *SRSF2* mutation, WEE1 inhibitor adavosertib increases the vulnerability of mutant cells to CHK1i, implying the potential of combined therapy for MDS/AML with *SRSF2* mutations [127].

### 4.3. Targeting Immune Responses and Inflammation Pathways

MDS is associated with immune dysfunction. On one hand, the activated innate immune system leads to abnormal hematopoiesis and unbalanced cell death and proliferation by directly affecting cytokine levels and activity of inflammatory pathways and immune cells. On the other hand, the adaptive immune system is activated by the expansion of malignant MDS stem cells, resulting in suppressed hematopoiesis and escaping from tumor surveillance. Due to this correlation, several immuno-suppressive and -modulatory therapies have been examined in selected MDS patients. Of note, LR-MDS has stronger inflammatory and cytotoxic features than HR-MDS, which is associated with a more suppressive microenvironment [82]. Moreover, the secretion profiles of cytokines are variable between different types of MDS; therefore, the treatment for each type of MDS, such as LR-MDS and HR-MDS, needs to be considered separately. For instance, the goal of treatment for LR-MDS is to correct peripheral cytopenia, whereas for patients with HR-MDS the primary target is the malignant clones [128].

A few small molecules targeting proinflammatory cytokines have been tested for the treatment of MDS patients, especially in LR-MDS. Luspatercept binds to TGF superfamily ligands and suppresses SMAD2/3 signaling. It was approved by FDA and EMA for transfusion-dependent LR-MDS patients with RS and/or *SF3B1* mutation after treatment with erythropoiesis-stimulating agents, and showed higher response rates [102,103,107,108,125,129]. For HR-MDS, *IRAK4-L* promotes leukemogenesis through NF-κB signaling [17,38] and a small molecule CA-4948 of oral IRAK4 inhibitor is currently evaluated in a phase 1 trial for HR-MDS and AML (NCT04278768) [106]. Results from 14 patients with *SF3B1*, *U2AF1*, or *FLT3* mutations were promising. Out of five AML patients, two reached complete remission/complete remission with partial hematologic recovery (CR/CRh) (1 CR, 1 CRh), and out of seven patients with spliceosome-mutated HR-MDS, four reached marrow CR [130]. The CA-4948 monotherapy was well tolerated and resulted in a reduction in bone marrow blasts in 89% of evaluated patients, and three patients with splicing factor mutations all achieved CR, and patients with objective responses also showed signs of hematologic recovery [131]. CA-4948 not only inhibits IRAK4 but also suppress FLT3-ITD AML progression in in vitro and in vivo models. It has shown safety and activity in patients with relapsed or refractory Non-Hodgkin Lymphomas and is now evaluated in patients with HR-MDS and AML [106].

In contrast, while many downstream splicing events have been reported to be altered due to splicing factor mutations in animal models and patient samples, including the important *EZH2* gene [4,9], no gene other than *IRAK4* has been shown to be a potential target for therapeutic purpose. Due to the complex immune responses in MDS/AML patients, further investigations should explore the immune response landscapes to search for specific targets. Clinical evaluations of individual treatments for LR-MDS and HR-MDS require careful consideration in future studies.

## 5. Concluding Remarks and Future Perspectives

Mutations in major splicing factors affect the expression of many downstream genes and elicit altered activity of multiple pathways and cellular responses. With an improved understanding of splicing defects, genome instability, and inflammatory responses, corresponding therapeutic strategies with small-molecule inhibitors have been developed targeting the spliceosome, ATR-CHK1 pathway, and related immune and inflammation responses (Figure 1). These new treatment options are either actively examined in labs or tested in clinical trials, holding new promise to substantiate our ability to fight against MDS/AML.

The treatment of leukemia with splicing modulators would encourage both basic and clinic research. On the basic research side, it provides an excellent study object for drug improvement with many basic questions to be addresses. How structurally disparate molecules target the same splicing factors, yet lead to different cellular outcomes? How does the interaction between drugs and target molecules occur and what is the mode of action? Furthermore, as currently used drugs target the core components of splicing devices, specificity and toxicity remain as major concerns. How can we increase the binding specificity of these small molecules and develop new drugs to specifically target individual splicing factors that are mutated in patients pose great challenges for chemical biology, too? On the clinical research side, pharmacological dynamics of the drug metabolism in vivo, toxicity and side effect to non-target organs and potential combinatory effects when splicing modulators are given with other drugs all need to be examined in the long run.

Genome instability is now appreciated as a potential mechanism for promoting the development of pre-leukemic HSCs and disease progression, especially with splicing factor mutations. Therefore, it underscores the importance to fully dissect the mechanism of DDR activation in HSCs/LSCs. The current consensus is that ATR-CHK1 signaling is activated specifically and implicated in maintaining the survival of mutant leukemia cell. With complete understanding of the DDR network, more targets will be identified and compared for clinical potential evaluation. Furthermore, as splicing factor mutations act through multiple mechanisms, it would be intriguing to test ATR/CHK1 inhibition in combination with inhibitors targeting other mechanisms, to see potential improvement of MDS/AML therapy in near future [132].

The traditional MDS/AML treatment methods by modulating immune response and pathways have been well studied. Ongoing research is anticipated to translate understanding of the complex molecular- and immune-pathophysiology of MDS/AML into the identification of new targets and new therapeutic options. Moreover, induced R-loop disruption and DNA damage may directly or indirectly contribute to the activation of innate immune responses in normal and/or malignant HSCs through specific mechanisms and factors [133]. All these results indicate that targeting DDR, and immune and inflammation pathways could be a potential strategy for better efficacy for MDS/AML patients harboring splicing factor mutations.

Different splicing factor mutations led to altered splicing of distinct groups of genes, and even the same mutation may affect the splicing of different gene targets in a context-dependent manner as well [15,134,135]. These observations have indeed made it difficult to search for common splicing targets for treatment. On the other hand, major splicing factor mutations all result in the aberrant activity of DNA damage repair pathways and immune responses, independent of splicing defects. Such new findings not only provide an explanation for different splicing factor mutations involved in similar hematological disorders, but also raise the possibility to explore a common target not in splicing regulation and regardless of the mutation type. Such direction will promote new therapeutical strategies to help patients to a broader extent. Moreover, it will be interesting to explore the connections between R-loops, DNA damage, and inflammation in MDS/AML, and test new treatment options that modulate combination of key targets for improved clinical benefits.

## Figures and Tables

**Figure 1 biomedicines-10-01972-f001:**
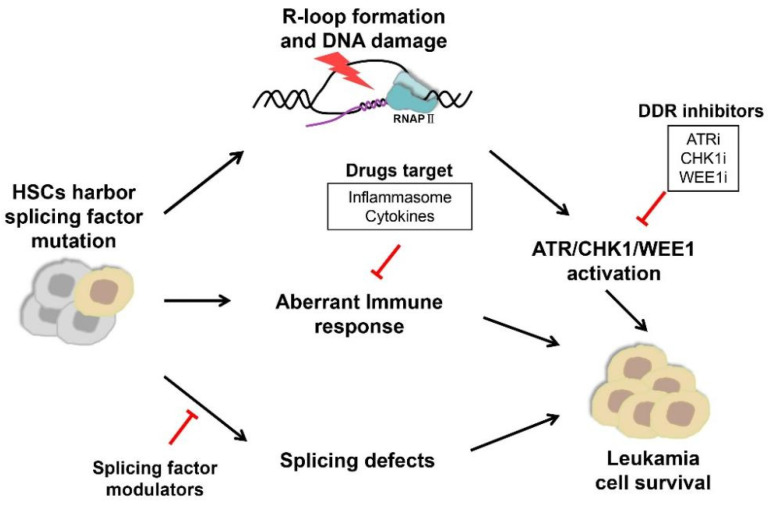
Approaches to targeting splicing factor mutations in splicing factor-mutant myelodysplastic syndromes/acute myeloid leukemia. Cells harboring splicing factor mutations have increased R loops and dysregulation of innate immune and inflammatory pathways. The elevated R-loop formation results in activation of the ATR signaling pathway and DNA-damage response. Leukemic cells harboring splicing factor mutations preferentially respond to ATR/CHK1/WEE1 inhibition and immune targeting agents (Inflammasome, Cytokines). In addition, cells with splicing factor mutations are more sensitive to splicing modulators that selectively inhibit SF3B1, RBM39, and arginine methyltransferase (PRMT) activity.

**Table 1 biomedicines-10-01972-t001:** Splicing factors involved in MDS/AML and their target genes.

Mutations	Pathway/Functions	Target Genes	Key References
*ZRSR2*	Transcription	*LZTR1*, *IRF7*, *E2F1*, *E2F2*, *E2F3*, *E2F4*, *E2F6*	[33,34,35]
	MAPK pathway	*MAPK1*, *MAPK3*, *MAPK9*	[35,36]
	Other	*RASGRP1, RASGRP2, RASGRP4, ARAF, BRAF, RAF1, PTEN*	[35]
*SF3B1*	MAPK pathway	*MAP3K7*	[10,37]
	Immune response or inflammation	*IRAK4*	[38]
	DNA damage	*ATR/CHK1*	[14]
	Other	*NF1*, *DICER1*, *PML*, *PDS5A*, *PPP2R5A*, *BRD9*	[37,39]
*U2AF1*	Epigenetic regulation	*H2AFY*, *ASXL1*, *BCOR*, *DNMT3B*	[11,31]
	Apoptosis	*CASP8*	[11,31]
	DNA damage	*ATR/CHK1*, *FANCA*	[11,31]
	Immune response or inflammation	*IRAK4*	[17]
	Other	*STRAP*, *CEP164*, *EHMT1*, *WAC*, *PABPC4*, *PPWD1*, *PTBP1*, *UPF3B*	[40]
*SRSF2*	Transcription	*EZH2*, *E2F1*	[9,41]
	Apoptosis	*CASP8*	[42]

**Table 2 biomedicines-10-01972-t002:** Summary of agents in pre-clinical and clinical for MDS/AML.

Category	Target	Agent	Pre-Clinical/Clinical Evaluation	Key References
Splicing factor modulators	SF3B complex	Pladienolides	Anti-tumor activities in various mouse xenograft models	[87]
		FR901464	Anti-tumor activities in mouse xenograft models	[87,88]
		Spliceostatin A		[87,88,92]
		Herboxidiene	Inhibit tumor growth in mouse xenograft models	[90]
		H3B-8800	NCT02841540/I/Recruiting	[3,93,94,95]
		E7107	Anti-tumor activities in mouse xenograft models	[12,96]
		Sudemycin D6		[91]
Other splicing modulators	RBM39	E7070	NCT01692197/II/Completed	[97]
	PRMT5	GSK3326595	NCT03614728/ I/Terminated	[98,99]
		JNJ-64619178	NCT03573310/I/Active	[98]
		PRT543	NCT03886831/I/Active	
DDR inhibitors	ATR	AZD6738	NCT03770429/I/Recruiting	[74,100]
		VE-822	Anti-tumor activities in various mouse xenograft models	[101,102]
		VE-821	Cellular/xenograft	[103]
	CHK1	UCN-01	NCT00301938/I/Completed	[14]
			NCT00004263/I/Completed	
		MK-8776 (SCH900776)	NCT00907517/I/Terminated	[104]
			NCT01870596/II/Completed	
		LY2603618	NCT02649764/I/Active	[105]
	WEE1	MK1775	Cellular	[103]
Inflammasome	IRAK4	CA-4948	NCT04278768/I/II/Active	[106]
Cytokines	TGF-β	Luspatercept	NCT02604433/III/Completed NCT02631070/III/Completed	[107] [107,108]

## Data Availability

Not applicable.

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
