# Peer review of "Molecular Threat of Splicing Factor Mutations to Myeloid Malignancies and Potential Therapeutic Modulations"

_biomedicines, 2022, doi:10.3390/biomedicines10081972_

Round 1

Reviewer 1 Report

The paper is interesting and well written. It is acceptable for publication

Author Response

Comment: The paper is interesting and well written. It is acceptable for publication.

Response: We appreciate the reviewer’s support for having our manuscript published in Biomedicines.

Reviewer 2 Report

The authors provide an excellent review of splicing factors mutated in MDS (primarily) and some of their targets. Less strong is the review of emergin therapies. This is the focus of the article. There nmeeds to be more discussion of what has been done, what hasn't worked, and what might work in targeting spliceosome mutations in MDS and related myeloid malignancies. Targets of splicing factors are not by themselves targets of myeloid malignancies with specific splicing factor mutations,. 

Author Response

Comment: The authors provide an excellent review of splicing factors mutated in MDS (primarily) and some of their targets. Less strong is the review of emerging therapies. This is the focus of the article. These needs to be more discussion of what has been done, what hasn't worked, and what might work in targeting spliceosome mutations in MDS and related myeloid malignancies. Targets of splicing factors are not by themselves targets of myeloid malignancies with specific splicing factor mutations. 

Response: We thank the reviewer for his/her constructive suggestion on providing more detailed information and thoughts regarding the emerging therapies. We thereby made modifications in the manuscript accordingly. More descriptions about the splicing factor modulators were added in lines 238-240, lines 243-245, and lines 273-277. More detailed information for targeting DDR with small molecule inhibitors was added in lines 299-306, lines 312-315 and lines 322-324. More information regarding targeting immune responses and inflammation pathways was incorporated in lines 359-363, lines 366-369, and lines 374-392. Taken together, with a more comprehensive review of emerging therapies, readers would obtain a clearer idea about the rationale and current evaluation status of various therapy molecules for myeloid malignancy treatment.

We agree with the reviewer’s comment that targets of splicing factors are not by themselves targets of myeloid malignancies with specific splicing factor mutations. To our knowledge, the SF3B1 K700E and U2AF1 mutations promote alternative splicing of IRAK4 to generate a long isoform (IRAK4-L) in MDS and AML, which interacts with MyD88 to elicit hyperactivation of NF-κB and leukemogenesis [1,2]. IRAK4 inhibitor CA-4948 is currently evaluated in a phase I trial for HR-MDS and AML (NCT04278768) [3]. Results from 14 patients with SF3B1, U2AF1, or FLT3 mutations were promising. Out of five AML patients with spliceosome mutations, two reached complete remission/complete remission with partial hematologic recovery (CR/CRh) (1 CR, 1 CRh), and out of seven patients with spliceosome-mutated HR-MDS, four reached marrow CR [4]. The CA-4948 monotherapy was well tolerated and resulted in a reduction in bone marrow blasts in 89% of evaluated patients. Three patients with spliceosome mutation all achieved CR, and patients with objective responses also showed signs of hematologic recovery [5]. CA-4948 not only inhibits IRAK4 but also suppresses FLT3-ITD AML in in vitro and in vivo models. Its safety and activity in patients with relapsed or refractory Non-Hodgkin Lymphomas were shown. CA-4948 is now evaluated in patients with high-risk MDS and AML, too [3]. To help the reader better understand the significance of IRAK4 splicing and potential of it inhibitor for treatment, we add the above statement to our revised manuscript from lines 376 to 388.

In contrast, while many downstream splicing events have been reported to be altered due to splicing factor mutations in animal models and patient samples, including the important EZH2 gene [6,7], no gene other than IRAK4 has been shown as a potential target for therapeutic purpose. Moreover, different splicing factor mutations alter the splicing of distinct groups of gene, making it difficult to search for the common splicing target for treatment, even though all splicing factor mutations seem to contribute to the progression of similar disease types. This is also the major reason we discuss non-splicing disease mechanisms and potential common targets modulating DDR and immune responses. We added this discussion into the Concluding Remarks and Future Perspectives part. To ensure the coherence and logic of the article, we have modified this part of the manuscript in lines 389-392.

Reference

  1. Obeng, E.A.; Chappell, R.J.; Seiler, M.; Chen, M.C.; Campagna, D.R.; Schmidt, P.J.; Schneider, R.K.; Lord, A.M.; Wang, L.; Gambe, R.G. Physiologic expression of Sf3b1K700E causes impaired erythropoiesis, aberrant splicing, and sensitivity to therapeutic spliceosome modulation. Cancer cell 2016, 30, 404-417.
  2. Choudhary, G.S.; Smith, M.A.; Pellagatti, A.; Bhagat, T.D.; Gordon, S.; Pandey, S.; Shah, N.; Aluri, S.; Booher, R.N.; Ramachandra, M. SF3B1 mutations induce oncogenic IRAK4 isoforms and activate targetable innate immune pathways in MDS and AML. Blood 2019, 134, 4224.
  3. Garcia-Manero, G.; Platzbecker, U.; Tarantolo, S.R.; Gropper, S.; Talati, C.; Götze, K.S.; Dugan, J.; Winer, E.S.; Martinez, E.; Lieberman, C. A Phase 1, Open Label Dose Escalation Trial Evaluating the Safety, Pharmacokinetics, Pharmacodynamics, and Clinical Activity of Orally Administered CA-4948 in Patients with Acute Myelogenous Leukemia or Myelodysplastic Syndrome. Blood 2020, 136, 16.
  4. Garcia-Manero, G.; Winer, E.S.; DeAngelo, D.J.; Tarantolo, S.R.; Sallman, D.A.; Dugan, J.; Groepper, S.; Giagounidis, A.; Gotze, K.S.; Metzeler, K. Phase 1/2a study of the IRAK4 inhibitor CA-4948 as monotherapy or in combination with azacitidine or venetoclax in patients with relapsed/refractory (R/R) acute myeloid leukemia or lyelodysplastic syndrome. 2022.
  5. Garcia‐Manero, G.; Tarantolo, S.; Verma, A.; Dugan, J.; Winer, E.; Giagounidis, A. A Phase 1, dose escalation trial with novel oral irak4 inhibitor ca‐4948 in patients with acute myelogenous leukemia or myelodysplastic syndrome–interim report. In Proceedings of the EHA Annual Meeting, 2021.
  6. Papaemmanuil, E.; Gerstung, M.; Malcovati, L.; Tauro, S.; Gundem, G.; Van Loo, P.; Yoon, C.J.; Ellis, P.; Wedge, D.C.; Pellagatti, A. Clinical and biological implications of driver mutations in myelodysplastic syndromes. Blood, The Journal of the American Society of Hematology 2013, 122, 3616-3627.
  7. Kim, E.; Ilagan, J.O.; Liang, Y.; Daubner, G.M.; Lee, S.C.-W.; Ramakrishnan, A.; Li, Y.; Chung, Y.R.; Micol, J.-B.; Murphy, M.E. SRSF2 mutations contribute to myelodysplasia by mutant-specific effects on exon recognition. Cancer cell 2015, 27, 617-630.

Reviewer 3 Report

The authors aim to review novel treatment possibilities for patients with hematological malignancies harboring splicing mutations.

Comments:

1) In the introduction it is mentioned that all splicing mutations are related to poor prognosis/progression but what about the SF3B1 mutations in MDS?

2) Although it is nice to speculate on potential additional pathways that may be targeted when splicing factor mutations are present, many of the treatments that are suggested do not select specifically this subgroup of patients. Moreover, the ultimate effect of splicing mutations are highly context dependent.

3) Already many references are included in the manuscript but most focus on the usage of the novel treatments and their mechanisms of action. Still, many recent papers on the role of splicing mutations and splicing modulation in hematological malignancies are missing, while according to the title it is expected to be the main focus.

Author Response

Comment:

1) In the introduction it is mentioned that all splicing mutations are related to poor prognosis/progression but what about the SF3B1 mutations in MDS?

Response: We apologize for not specifying the prognosis relevance of SF3B1 mutations in MDS in our last submission. Mutations in SF3B1 are detected across various types of blood and solid tumors, such as CLL [1], and uveal melanoma (UM) [2]. As the frequency of SF3B1 mutations is highest in MDS, particularly in MDS with ring sideroblasts (MDS-RS) [3,4], they are now used to define a distinct clinical entity of MDS-RS [5].

      Early studies suggested that MDS with SF3B1 mutations were associated with a better prognosis and a lower likelihood to transform into AML compared with MDS free of SF3B1 mutation [6-10]. MDS and UM patients carrying SF3B1 mutations appeared to have better survival as well [11,12]. In contrast, these conclusions were not reproduced in some other studies [13,14], especially when patients harbor other gene mutations [6]. SF3B1 mutations in UM was later linked to higher risk of metastasis, too [15]. Interestingly, the SF3B1 K700E mutation promotes alternative splicing of IRAK4 to generate a long isoform (IRAK4-L) in MDS and AML, which interacts with MyD88 to elicit hyperactivation of NF-κB and leukemogenesis [16,17]. Clinically, the expression of IRAK-L is associated with poor prognosis of MDS and AML [18]. Other tumorigenic mechanisms, including MYC stabilization, were also reported [19,20]. Thus, the prognostic relevance of SF3B1 mutations in MDS and other diseases may depend on cellular contexts and await further confirmation. We have incorporated these reports and discussion into the revised manuscript in lines 88-101.

2) Although it is nice to speculate on potential additional pathways that may be targeted when splicing factor mutations are present, many of the treatments that are suggested do not select specifically this subgroup of patients. Moreover, the ultimate effects of splicing mutations are highly context dependent.

Response: We appreciate the reviewer’s comments and agree that DDR and immune regulatory pathways could be potential targets for patients with blood cancer or other types of tumors, who do not necessarily harbor splicing factor mutations. On the other hand, a key question our paper would like to discuss about is whether splicing factor mutations lead to specifically elevated DNA damage risk and immune responses, therefore forcing the mutant tumor cells to more rely on DDR activity or certain immune signaling pathway for survival. According to accumulating evidence from recent studies, the answer might be Yes.

     A number of reports showed that mutations in SF3B1, U2AF1, and SRSF2 all cause genome instability via R-loop independent of splicing defects, resulting in higher sensitivity of mutant cells to DDR inhibitors than cells without splicing factor mutations [21-23]. Recent studies also showed that mutations in SF3B1, U2AF1, and SRSF2 seem to alter the activity of innate immune pathways and enhance chronic inflammation [18,24]. Moreover, the common effects of these mutant splicing factors on NF-κB signaling have raised the possibility that the increased activation of NF-κB may be a convergent mechanism to promote disease [18,25,26]. All these results indicate that targeting DDR, and immune and inflammation pathways could be a potential strategy for better efficacy for MDS/AML patients harboring splicing factor mutations. Moreover, researchers have also examined the potential therapeutical effects of DDR inhibitor in combination with splicing factor modulators (e.g. E7107), and certain cell and pre-clinical studies suggested that combined treatment may increase the efficacy of splicing factor modulators for tumor cells with splicing factor mutations [27].

      Regarding the context-dependent effects of splicing mutations, we agree with the reviewer and have learned from previous studies by us and others that different splicing factor mutations led to altered splicing of distinct groups of genes, and even the same mutation may affect the splicing of different gene targets in a context-dependent manner as well [28,29]. These observations have indeed made it more difficult to search for splicing targets for treatment. On the other hand, as we mentioned above, major splicing factor mutations result in the aberrant activity of DNA damage and immune response, which is exactly consistent with the fact that different splicing factor mutations all cause similar hematological abnormality. Moreover, these findings of alternative and shared disease mechanism raise the possibility to find a common target for therapy, regardless of the mutation type. Taken together, while splicing factor mutations have distinct effects on many aspects of cell functions, the exact purpose of this review is to summarize our current understanding of common disease mechanisms related to different splicing mutations, which would elicit the development of drugs that would help patients to a broader extent. We have added this part of information and discussion into the revised manuscript in lines 445-457.

3) Already many references are included in the manuscript but most focus on the usage of the novel treatments and their mechanisms of action. Still, many recent papers on the role of splicing mutations and splicing modulation in hematological malignancies are missing, while according to the title it is expected to be the main focus.

Response: We apologize for not having all papers about the roles of splicing mutations and splicing modulation in hematological malignancies included in our last. As the reviewer suggested, we added more discussion about SF3B1 mutations in lines 88-101, and included more reports and descriptions about mutations of other splicing factor and snRNAs in lines 161-172. In case of any report missed in our review, we expressed our apology in the Acknowledge part of the revised manuscript.

Reference

  1. Seiler, M.; Peng, S.; Agrawal, A.A.; Palacino, J.; Teng, T.; Zhu, P.; Smith, P.G.; Caesar-Johnson, S.J.; Demchok, J.A.; Felau, I. Somatic mutational landscape of splicing factor genes and their functional consequences across 33 cancer types. Cell reports 2018, 23, 282-296. e284.
  2. Gallenga, C.E.; Franco, E.; Adamo, G.G.; Violanti, S.S.; Tassinari, P.; Tognon, M.; Perri, P. Genetic Basis and Molecular Mechanisms of Uveal Melanoma Metastasis: A Focus on Prognosis. Frontiers in Oncology 2022, 12, 828112-828112.
  3. Saez, B.; Walter, M.J.; Graubert, T.A. Splicing factor gene mutations in hematologic malignancies. Blood, The Journal of the American Society of Hematology 2017, 129, 1260-1269.
  4. Pagliuca, S.; Gurnari, C.; Visconte, V. Molecular targeted therapy in myelodysplastic syndromes: new options for tailored treatments. Cancers 2021, 13, 784.
  5. Yoshida, K.; Sanada, M.; Shiraishi, Y.; Nowak, D.; Nagata, Y.; Yamamoto, R.; Sato, Y.; Sato-Otsubo, A.; Kon, A.; Nagasaki, M. Frequent pathway mutations of splicing machinery in myelodysplasia. Nature 2011, 478, 64-69.
  6. Cazzola, M.; Rossi, M.; Malcovati, L. Biologic and clinical significance of somatic mutations of SF3B1 in myeloid and lymphoid neoplasms. Blood, The Journal of the American Society of Hematology 2013, 121, 260-269.
  7. Makishima, H.; Visconte, V.; Sakaguchi, H.; Jankowska, A.M.; Abu Kar, S.; Jerez, A.; Przychodzen, B.; Bupathi, M.; Guinta, K.; Afable, M.G. Mutations in the spliceosome machinery, a novel and ubiquitous pathway in leukemogenesis. Blood, The Journal of the American Society of Hematology 2012, 119, 3203-3210.
  8. Seo, J.Y.; Lee, K.-O.; Kim, S.-H.; Kim, K.; Jung, C.W.; Jang, J.H.; Kim, H.-J. Clinical significance of SF3B1 mutations in Korean patients with myelodysplastic syndromes and myelodysplasia/myeloproliferative neoplasms with ring sideroblasts. Annals of hematology 2014, 93, 603-608.
  9. Patnaik, M.M.; Lasho, T.L.; Hodnefield, J.M.; Knudson, R.A.; Ketterling, R.P.; Garcia-Manero, G.; Steensma, D.P.; Pardanani, A.; Hanson, C.A.; Tefferi, A. SF3B1 mutations are prevalent in myelodysplastic syndromes with ring sideroblasts but do not hold independent prognostic value. Blood, The Journal of the American Society of Hematology 2012, 119, 569-572.
  10. Hosono, N. Genetic abnormalities and pathophysiology of MDS. International journal of clinical oncology 2019, 24, 885-892.
  11. Harbour, J.W.; Roberson, E.; Anbunathan, H.; Onken, M.D.; Worley, L.A.; Bowcock, A.M. Recurrent mutations at codon 625 of the splicing factor SF3B1 in uveal melanoma. Nature genetics 2013, 45, 133-135.
  12. Quek, C.; Rawson, R.V.; Ferguson, P.M.; Shang, P.; Silva, I.; Saw, R.P.; Shannon, K.; Thompson, J.F.; Hayward, N.K.; Long, G.V. Recurrent hotspot SF3B1 mutations at codon 625 in vulvovaginal mucosal melanoma identified in a study of 27 Australian mucosal melanomas. Oncotarget 2019, 10, 930.
  13. Damm, F.; Kosmider, O.; Gelsi-Boyer, V.; Renneville, A.; Carbuccia, N.; Hidalgo-Curtis, C.; Della Valle, V.; Couronné, L.; Scourzic, L.; Chesnais, V. Mutations affecting mRNA splicing define distinct clinical phenotypes and correlate with patient outcome in myelodysplastic syndromes. Blood, The Journal of the American Society of Hematology 2012, 119, 3211-3218.
  14. Thol, F.; Kade, S.; Schlarmann, C.; Löffeld, P.; Morgan, M.; Krauter, J.; Wlodarski, M.W.; Kölking, B.; Wichmann, M.; Görlich, K. Frequency and prognostic impact of mutations in SRSF2, U2AF1, and ZRSR2 in patients with myelodysplastic syndromes. Blood, The Journal of the American Society of Hematology 2012, 119, 3578-3584.
  15. Yavuzyigitoglu, S.; Koopmans, A.E.; Verdijk, R.M.; Vaarwater, J.; Eussen, B.; Van Bodegom, A.; Paridaens, D.; Kiliç, E.; de Klein, A.; Group, R.O.M.S. Uveal melanomas with SF3B1 mutations: a distinct subclass associated with late-onset metastases. Ophthalmology 2016, 123, 1118-1128.
  16. Obeng, E.A.; Chappell, R.J.; Seiler, M.; Chen, M.C.; Campagna, D.R.; Schmidt, P.J.; Schneider, R.K.; Lord, A.M.; Wang, L.; Gambe, R.G. Physiologic expression of Sf3b1K700E causes impaired erythropoiesis, aberrant splicing, and sensitivity to therapeutic spliceosome modulation. Cancer cell 2016, 30, 404-417.
  17. Choudhary, G.S.; Smith, M.A.; Pellagatti, A.; Bhagat, T.D.; Gordon, S.; Pandey, S.; Shah, N.; Aluri, S.; Booher, R.N.; Ramachandra, M. SF3B1 mutations induce oncogenic IRAK4 isoforms and activate targetable innate immune pathways in MDS and AML. Blood 2019, 134, 4224.
  18. Smith, M.A.; Choudhary, G.S.; Pellagatti, A.; Choi, K.; Bolanos, L.C.; Bhagat, T.D.; Gordon-Mitchell, S.; Von Ahrens, D.; Pradhan, K.; Steeples, V. U2AF1 mutations induce oncogenic IRAK4 isoforms and activate innate immune pathways in myeloid malignancies. Nature cell biology 2019, 21, 640-650.
  19. Darman, R.B.; Seiler, M.; Agrawal, A.A.; Lim, K.H.; Peng, S.; Aird, D.; Bailey, S.L.; Bhavsar, E.B.; Chan, B.; Colla, S. Cancer-associated SF3B1 hotspot mutations induce cryptic 3′ splice site selection through use of a different branch point. Cell reports 2015, 13, 1033-1045.
  20. Liu, Z.; Yoshimi, A.; Wang, J.; Cho, H.; Lee, S.C.-W.; Ki, M.; Bitner, L.; Chu, T.; Shah, H.; Liu, B. Mutations in the RNA splicing factor SF3B1 promote tumorigenesis through MYC stabilization. Cancer discovery 2020, 10, 806-821.
  21. Singh, S.; Ahmed, D.; Dolatshad, H.; Tatwavedi, D.; Schulze, U.; Sanchi, A.; Ryley, S.; Dhir, A.; Carpenter, L.; Watt, S.M. SF3B1 mutations induce R-loop accumulation and DNA damage in MDS and leukemia cells with therapeutic implications. Leukemia 2020, 34, 2525-2530.
  22. Chen, L.; Chen, J.-Y.; Huang, Y.-J.; Gu, Y.; Qiu, J.; Qian, H.; Shao, C.; Zhang, X.; Hu, J.; Li, H. The augmented R-loop is a unifying mechanism for myelodysplastic syndromes induced by high-risk splicing factor mutations. Molecular cell 2018, 69, 412-425. e416.
  23. Nguyen, H.D.; Yadav, T.; Giri, S.; Saez, B.; Graubert, T.A.; Zou, L. Functions of replication protein A as a sensor of R loops and a regulator of RNaseH1. Molecular cell 2017, 65, 832-847. e834.
  24. Pollyea, D.A.; Harris, C.; Rabe, J.L.; Hedin, B.R.; De Arras, L.; Katz, S.; Wheeler, E.; Bejar, R.; Walter, M.J.; Jordan, C.T. Myelodysplastic syndrome-associated spliceosome gene mutations enhance innate immune signaling. Haematologica 2019, 104, e388.
  25. Lee, S.C.-W.; North, K.; Kim, E.; Jang, E.; Obeng, E.; Lu, S.X.; Liu, B.; Inoue, D.; Yoshimi, A.; Ki, M. Synthetic lethal and convergent biological effects of cancer-associated spliceosomal gene mutations. Cancer cell 2018, 34, 225-241. e228.
  26. Ramos, N.R.; Mo, C.C.; Karp, J.E.; Hourigan, C.S. Current approaches in the treatment of relapsed and refractory acute myeloid leukemia. Journal of clinical medicine 2015, 4, 665-695.
  27. Nguyen, H.D.; Leong, W.Y.; Li, W.; Reddy, P.N.; Sullivan, J.D.; Walter, M.J.; Zou, L.; Graubert, T.A. Spliceosome mutations induce R loop-associated sensitivity to ATR inhibition in myelodysplastic syndromes. Cancer research 2018, 78, 5363-5374.
  28. Dvinge, H. Regulation of alternative mRNA splicing: old players and new perspectives. FEBS letters 2018, 592, 2987-3006.
  29. Bejar, R. Splicing factor mutations in cancer. In RNA processing; Springer: 2016; pp. 215-228.

Round 2

Reviewer 2 Report

this is stronger and more focused. 

Author Response

Comment: This is stronger and more focused.

Response: We appreciate the reviewer’s approval for the publication of our manuscript.

Reviewer 3 Report

I thank the authors for their reply. Still, I think the title is not completely covering the contents of the manuscript. In my view, the presented treatment options are not "emerging" treatments but more "potential" treatments for splicing factor mutated cancers based on proposed downstream pathway alterations. A minor point is that the additional papers they suggest are not specifically based on data present on hematological malignancies with splicing factor mutations (pre)clinically treated with novel agents. This is also a little confusing in the title of table 2 where trials are presented that are performed irrespective of splicing factor mutations. 

Author Response

Comment:

I thank the authors for their reply. Still, I think the title is not completely covering the contents of the manuscript. In my view, the presented treatment options are not "emerging" treatments but more "potential" treatments for splicing factor mutated cancers based on proposed downstream pathway alterations.

Response: We thank the review for his/her excellent suggestion. We agree that the disease mechanism with splicing factor mutations is ongoing and most of the related new treatment tests are still at the early stage. To cover our discussion on both disease mechanisms and therapeutic examination, we changed the title to “Molecular Threat of Splicing Factor Mutations to Myeloid Malignancies and Potential Therapeutic Modulations” accordingly.

A minor point is that the additional papers they suggest are not specifically based on data present on hematological malignancies with splicing factor mutations (pre)clinically treated with novel agents. This is also a little confusing in the title of table 2 where trials are presented that are performed irrespective of splicing factor mutations. 

Response: We appreciate the reviewer’s careful examination of the newly added papers. According, we deleted some papers and clinical studies, in which the drugs were not directly tested in cancers with splicing factor mutations. Specifically, for molecule inhibitors targeting DDR, we deleted the description about AZ20 (ref. 111) and GDC-0575 (ref. 118-119). For drugs targeting immune responses and inflammation pathways, we deleted contents regarding clinical trials with Canakinumab (NCT04810611, NCT04239157, and NCT04798339), Etanercept (NCT00118287), ibrutinib (NCT02553941 and NCT03359460), OPN-305 (NCT02363491), bortezomib (NCT00580242), Cx-01 (NCT02995655), and OPN-305 (NCT02363491). Corresponding references that were deleted included 129-131 and 136, 138-141. After these modifications, the total number of cited papers was changed from 147 to 136. On the other hand, we kept some other papers, including references 93-94 that evaluated the toxicity of E7107, and reference 96 that showed H3B-8800 was safe even with prolonged dosing in myeloid neoplasms.

For table 2, we listed recent clinical trials with agents targeting molecular pathways that were shown to be aberrantly altered in hematologic malignancies by splicing factor mutations. While these clinical trials did not all specifically focus on splicing factor mutations, the results of which may help readers to evaluate the potential of these small molecules in treating patients with splicing factor mutations. According to the reviewer’s comment, we modified the title of table 2 to avoid any misunderstanding.
